# Learning from True-False Labels via Multi-modal Prompt Retrieving

**Zhongnian Li** [1 2]  **Jinghao Xu** [1]  **Peng Ying** [1]  **Meng Wei** [1]  **Xinzheng Xu** [1 2 3]

## Abstract

Pre-trained **V**ision-**L**anguage **M**odels (VLMs) exhibit strong zero-shot classification abilities, demonstrating great potential for generating weakly supervised labels. Unfortunately, existing weakly supervised learning methods are short of ability in generating accurate labels via VLMs. In this paper, we propose a novel weakly supervised labeling setting, namely **T**rue-**F**alse **L**abels (TFLs) which can achieve high accuracy when generated by VLMs. The TFL indicates whether an instance belongs to the label, which is randomly and uniformly sampled from the candidate label set. Specifically, we theoretically derive a risk-consistent estimator to explore and utilize the conditional probability distribution information of TFLs. Besides, we propose a convolutional-based **M**ulti-modal **P**rompt **R**etrieving (MRP) method to bridge the gap between the knowledge of VLMs and target learning tasks. Experimental results demonstrate the effectiveness of the proposed TFL setting and MRP learning method. The code to reproduce the experiments is at github.com/Tranquilxu/TMP.

## 1. Introduction

In recent years, supervised learning has exhibited remarkable performance across a diverse range of visual tasks, including image classification(Dosovitskiy et al., 2021), object detection(Zong et al., 2023), and semantic segmentation(Chen et al., 2018). This success can be largely attributed to the abundance of fully annotated training web data on the internet. However, a significant challenge remains in the time-consuming process of collecting such annotated datasets. To address this challenge, various forms of weakly supervised learning have been proposed and explored in a range of settings, including semi-supervised learning(Van Engelen & Hoos, 2020; Cao et al., 2022; Guo et al., 2022; Wei et al., 2024), positive-unlabeled learning(Sakai et al., 2018; Chapel et al., 2020; Hu et al., 2021; Li et al., 2024b; 2025), noisy-label learning(Menon et al., 2015; Ghosh et al., 2017; Han et al., 2020), partial-label learning(Cour et al., 2011a; Feng et al., 2020b; Xia et al., 2023; Li et al., 2024a), and complementary-label learning(Ishida et al., 2017; Yu et al., 2018; Gao & Zhang, 2021; Wei et al., 2023).

Recently, pre-trained **V**ision-**L**anguage **M**odel**s** (VLMs) (Radford et al., 2021; Li et al., 2022; Liu et al., 2024) trained on large-scale labeled data have achieved remarkable results and exhibit significant potential in generating high-quality weakly supervised labels, thereby reducing annotation costs(Menghini et al., 2023). Unfortunately, as shown in Figure 1 (a), the VLMs supervised labels are often of low quality because the zero-shot results from VLMs may be incorrect(Wang et al., 2022; Menghini et al., 2023). Specifically, the VLMs supervised label will be annotate as "Sea horse" (i.e., label with the highest confidence), while the ground-truth label is "Bass". These VLMs supervised labels with noise semantics may degrade the performance of models on target learning tasks. This fact further inspires us to explore and leverage novel weakly supervised labeling settings to harness VLMs for generating higher-quality labels.

In this paper, we propose a novel weakly supervised classification setting: learning from **True-False Labels** (TFLs), which can achieve high accuracy and avoid the cost of manual annotation when generated by VLMs. Besides, the utilization of TFLs can markedly enhance the efficiency of human annotation. The TFL indicates *whether an instance belongs to the label*, which is randomly and uniformly sampled from the candidate label set. Specifically, an instance will be annotated with a "True" label when it belongs to the sampled label, and with a "False" label when it does not. For example, as illustrated in Figure 1 (b), for an image with the ground-truth label "Bass", annotators will easily annotate the instance with "Bass" and "True" label when the randomly sampled label is also "Bass". Conversely, the

---
[1]School of Computer Science and Technology, China University of Mining and Technology, Xuzhou, China [2]Mine Digitization Engineering Research Center of the Ministry of Education, China University of Mining and Technology, Xuzhou, China [3]State Key Lab. for Novel Software Technology, Nanjing University, Nanjing, China. Correspondence to: Xinzheng Xu <xxzheng@cumt.edu.cn>.

*Proceedings of the 42nd International Conference on Machine Learning*, Vancouver, Canada. PMLR 267, 2025. Copyright 2025 by the author(s).

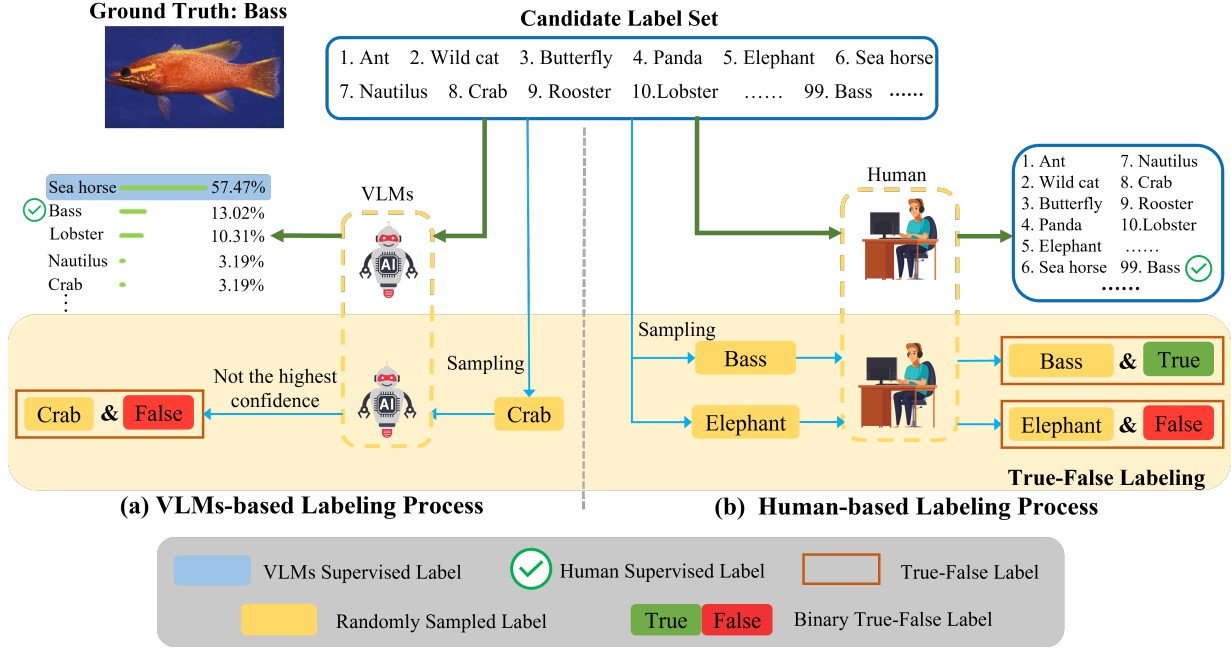

*Figure 1.* A comparison between traditional labeling and TF labeling (both VLMs-based and human-based labeling process). The VLMs zero-shot result depicted in the figure are derived from CLIP with ViT-L/14(Radford et al., 2021). The example images and categories are derived from Caltech-101(Fei-Fei et al., 2004). True-False labels can enhance both the accuracy of VLMs supervised labeling and the efficiency of Human-based labeling.

annotators can annotate the instance with "Elephant" and "False" label when the randomly sampled label is "Elephant". Moreover, as illustrated in Figure 1 (a), when the zero-shot results from VLMs are incorrect, the TFLs generated by VLMs are almost always accurate. Overall, this novel labeling setting can effectively leverage the knowledge of VLMs for generating high-quality labels. Additionally, the TFLs can enhance the efficiency of the human labeling process by reducing the time cost for browsing the candidate label set.

In this paper, we propose a risk-consistent method to learn from **T**rue-False labels via **M**ulti-modal **P**rompt retrieving (TMP). Specifically, we theoretically derive a risk-consistent estimator to explore and utilize the conditional probability distribution information of TFLs instead of relying solely on labels. Besides, we introduce a novel prompt learning method called MRP learning, which can bridge the gap between pre-training and target learning tasks. Extensive experiments on various datasets clearly demonstrate the effectiveness of the proposed TFL setting and MPR learning method.

Our main contributions are summarized as follows:

- We propose a novel labeling setting for weakly supervised classification, which can effectively leverage the knowledge of VLMs for generating high-quality labels and enhance the efficiency of the human labeling process.

- A risk-consistent method is introduced to explore and utilize the conditional probability distribution information of TFLs instead of relying solely on labels. The conditional probability distribution information can be easily obtained by VLMs.

- A convolutional-based multi-modal prompt retrieving method is proposed to bridge the gap between the knowledge of VLMs and target learning tasks. To the best of our knowledge, this is the first convolutional-based prompt learning approach for weakly supervised learning.

## 2. Related Work

### 2.1. Weakly Supervised Learning

Weakly supervised learning aims to construct predictive models by learning from a large number of training samples that contain incomplete, inexact, or inaccurate supervision information(Zhou, 2018). These weakly supervised learning approaches include but not limited to semi-supervised learning(Van Engelen & Hoos, 2020; Cao et al., 2022; Guo et al., 2022; Wei et al., 2024), partial-label learning(Sakai et al., 2018; Chapel et al., 2020; Hu et al., 2021; Li et al., 2024b; 2025) and complementary-label learning(Ishida et al., 2017; Yu et al., 2018; Gao & Zhang, 2021; Wei et al., 2023).

Semi-supervised learning assumes the presence of both labeled and unlabeled data in the training set. It mainly includes entropy minimization methods(Grandvalet & Bengio, 2004; Lee et al., 2013), consistency regularization methods(Sajjadi et al., 2016; Tarvainen & Valpola, 2017; Miyato et al., 2018), and holistic methods(Berthelot et al., 2019; Sohn et al., 2020; Sosea & Caragea, 2023). Partial-label learning involves training instances with a set of potential candidate labels, where only one is assumed to be correct but is unknown. This approach can be categorized into identification-based strategies(Zhang et al., 2016; Tang & Zhang, 2017; Xu et al., 2019) and average-based strategies(Cour et al., 2011b; Hüllermeier & Beringer, 2005), depending on how they handle candidate labels. Complementary-label learning (CLL) assigns a label which specifies the class that an instance does not belong to. Ishida et al.(Ishida et al., 2017) design an unbiased risk estimator (URE) with a solid theoretical analysis, which enables multi-class classification with only complementary labels. Subsequently, various models and loss functions are incorporated into the CLL framework(Yu et al., 2018; Gao & Zhang, 2021; Ishida et al., 2019).

Recent studies have explored the potential of reducing annotation costs with weakly supervised learning. Unfortunately, these methods struggle to leverage the knowledge of VLMs to generate usable labels. Consequently, we propose the True-False label setting, which can achieve high-accuracy when generated by VLMs.

## 2.2. Prompt Learning in VLMs

The role of the prompt is primarily to provide the model with context and parameter information about the input. Prompts can help the model understand the input's intention and generate an appropriate response(Liu et al., 2023; Lei et al., 2024).

CLIP(Radford et al., 2021) introduces prompt to the CV and multi-modal domains by converting image category labels into text sequences as a hand-crafted language template prompt, such as "a photo of a {CLASS}". CoOp(Zhou et al., 2022b) transforms CLIP's hand-crafted template prompts into a set of learnable continuous vectors, which are optimized from few-shot transfer. CoCoOp(Zhou et al., 2022a) enhances CoOp by training a lightweight neural network to generate input conditional vectors for each image, resulting in better performance on new classes. VPT(Jia et al., 2022) introduces a small number of trainable parameters into the input space while keeping the pre-trained Transformer backbone frozen. These additional parameters are simply prepended into the input sequence of each Transformer layer and learned together with a linear head during fine-tuning. MaPLe(Khattak et al., 2023) develops a multi-modal prompt to improve consistency between visual and language representations.

Previous prompt learning approaches have typically focused on directly learning the prompt itself. In contrast, our method involves training a convolutional neural network to retrieve the prompt embeddings.

## 3. Method

In this section, we provide a detailed description of a risk-consistent method to learn from **T**rue-False labels via **M**ulti-modal **P**rompt retrieving (TMP). Firstly, we introduce the problem definition and the labeling process of TFL. Besides, we theoretically derive a risk-consistent estimator to explore and utilize the conditional probability distribution information of TFLs instead of relying solely on labels. Subsequently, we propose a convolutional-based multi-modal prompt retrieving method to bridge the gap between the knowledge of VLMs and target learning tasks. Finally, we illustrate the architecture of TMP.

### 3.1. True-False Labels

In contrast to the previous approach, we now consider another scenario, namely **T**rue-**F**alse **L**abels (TFLs) learning. In this setting, the TFL indicates whether an instance belongs to the label, which is randomly and uniformly sampled from the candidate label set. Specifically, annotator only needs to provide the binary TFL (i.e., "True" or "False") according to the randomly sampled label. To illustrate, as shown in Figure 1 (b), when considering the candidate label set, {"ant", "wild cat", "butterfly", "panda", "elephant", "sea horse", ⋯ "bass", ⋯}, for a "bass" image, the annotator can readily assign a "True" label when the randomly sampled label is the "bass". In contrast, the annotator can readily assign a "False" label when the randomly sampled label is the "elephant". Moreover, as illustrated in Figure 1 (a), the zero-shot results with the highest confidence from the VLMs annotate the instance with "sea horse". However, when the randomly sampled label is "crab", the TFL of this instance will be "crab" and "false", which is accurate. Compared to the traditional labeling, TFL effectively leverage the knowledge of VLMs for generating high-quality labels. Additionally, the TFLs enhance the efficiency of the human labeling process by reducing the time cost for browsing the candidate label set.

**Accuracy superiority.** As shown in Table 1, we demonstrate the accuracy superiority of TFL over traditional labeling. When utilizing CLIP for annotation, CLIP provides the binary TFL by determining whether the CLIP zero-shot result are consistent with the randomly sampled label. TFLs generally demonstrate an impressive accuracy rate exceeding 99.5%, which substantiates the effectiveness of TFL. Moreover, it is worth noting that, according to previous

*Table 1.* The TFL annotation details of five benchmark datasets. We demonstrate the accuracy of TFL by utilizing VLMs (i.e., CLIP ViT-L/14(Radford et al., 2021)) for annotation. We show the number of true labels and the efficiency of TFL by human annotation. # denotes the number of, and $\times$ denotes times.

| | Basic Information | | | Accuracy | | Efficiency |
|---|---|---|---|---|---|---|
| | # Classes | # Training Set | # True Labels | # Mislabeled | Error Rate (%) | Labeling Speed Multiplier $v_m$ ($\times$) |
| CIFAR-100 | 100 | 50000 | 498 | 256 | 0.512 | 50 |
| Tiny ImageNet | 200 | 100000 | 507 | 267 | 0.267 | 100 |
| Caltech-101 | 102 | 6400 | 65 | 18 | 0.281 | 51 |
| Food-101 | 101 | 75750 | 795 | 153 | 0.202 | 50.5 |
| Stanford Cars | 196 | 8144 | 42 | 20 | 0.246 | 98 |
| Average | - | - | - | 142.8 | 0.302 | 69.9 |

studies(Northcutt et al., 2021), TFLs generated by VLMs have, to some extent, achieved superior accuracy compared to traditional manual labeling.

**Efficiency superiority.** As shown in Table 1, we demonstrate the efficiency superiority of TFL over traditional labeling. Specifically, we introduce a labeling speed multiplier $v_m$ to quantify the efficiency of TFL. We define the time it takes for an annotator to determine whether an instance belongs to a label as the unit time. Thus, the time required for TFL to label an instance is 1 unit (i.e., $t_{TF} = 1$). In contrast, traditional human-based labeling process require annotators to browse through half of the candidate label set on average, so the mathematic expectation of traditional labeling time $t_c$ is $\frac{K}{2}$, where $K$ is the number of candidate labels. Then $v_m$ can be formulated as $v_m = \frac{t_c}{t_{TF}}$. As shown in Table 1, TFL demonstrates an average labeling efficiency that is 69.9 times higher than traditional labeling across five datasets, which substantiates the efficiency of TFL. The efficiency of TFLs can be attributed to the fact that this labeling approach significantly reduces the time spent browsing the candidate label set.

**Key advantages.** The TFL framework introduces three key advantages that fundamentally address limitations in standard self-training methods : ❶ **Random label sampling as implicit regularization:** TFL employs uniform random label sampling as implicit regularization to prevent model overfitting. This stochastic mechanism in TFL, analogous to established techniques like dropout, SGD and random data augmentation, promotes better generalization solutions(Ali et al., 2020; Wang et al., 2024). Unlike confidence-based pseudo-labeling in self-training methods , which amplifies CLIP's inherent biases through error propagation(Wang et al., 2022; Menghini et al., 2023). ❷ **High-accuracy TFL for bias amplification mitigation:** Prior studies(Wang et al., 2022; Menghini et al., 2023) have demonstrated that pseudo-labeling noise can create a cumulative effect on classes with inaccurate pseudo-labels, which progressively amplifies the model's inherent bias toward certain classes. This observation directly motivates TFL's design objective

to mitigate bias amplification through stochastic sampling, since such error propagation fundamentally stems from deterministic labeling mechanisms. Through its implementation, TFL achieves over 99% annotation accuracy, a result that significantly improves labeling precision via confidence-ranking-independent label generation. This advancement substantially reduces noise propagation by effectively minimizing error accumulation in pseudo-labeling iterations. ❸ **Hybrid supervision mechanism:** TFL integrates strong supervision from retained true labels to provide semantic correction anchors. This hybrid labels provide explicit optimization direction, enhances noise robustness and improves stability in complex scenarios.

### 3.2. Problem Setup

In multi-class classification, let $\mathcal{X} \in \mathbb{R}^d$ be the feature space and $\mathcal{Y} = [K]$ be the label space, where d is the feature space dimension; $[K] = \{1, \cdots, K\}$; and $K > 2$ is the number of classes. Suppose $D = \{(x_l, y_l)\}_{l=1}^N$ is the dateset where $x_l \in \mathcal{X}$ , $y_l \in \mathcal{Y}$ and $N$ denotes the number of training instances. We assume that $\{(x_l, y_l)\}_{l=1}^N$ are sampled independently from an unknown probability distribution with density $p(x, y)$. The goal of ordinary multi-class classification is to learn a classifier $f(x) : x \to \{1, \ldots, K\}$ that minimizes the classification risk with multi-class loss $\mathcal{L}(f(x), y)$ :

$$
\begin{aligned}
R(f) &= \mathbb{E}_{p(x,y)} \mathcal{L}(f(x), y) \\
&= \mathbb{E}_{x \sim \mu} \sum_{i=1}^K p(y = i|x) \mathcal{L}(f(x), i)
\end{aligned}
\tag{1}
$$

where $\mathbb{E}$ denotes the expectation.

In this paper, we consider the scenario where each instance is annotated with a TFL $Y$ instead of a ordinary class label $y$. Suppose the TF labeled training dataset $D_{TF} = \{(x_l, Y_l)\}_{l=1}^N$ is sampled randomly and uniformly from an unknown probability distribution with density $p(x, Y)$. $Y_l = (\bar{y}_l, s_l)$ is a TFL where $\bar{y}_l \in \mathcal{Y}$ is the randomly sampled label and $s_l \in \{0, 1\}$ represents whether instance $x_l$ belongs to category $\bar{y}_l$. Specifically, $s_l = 0$ signifies that the instance $x_l$ does not belong to the cate-

gory $\bar{y}_l$ and $s_l = 1$ denotes that the instance $x_l$ belongs to the category $\bar{y}_l$. Similarly, the objective is to learn a classifier $f(x) : x \rightarrow \{1, \ldots, K\}$ from the TF labeled training dataset, which can accurately categorize images that have not been previously observed.

### 3.3. Risk-Consistent Estimator

In this section, based on proposed problem setup, we present a risk-consistent method(Feng et al., 2020c;a; Xu et al., 2022). To rigorously depict the connection between ground-truth label and TFL, we introduce the following assumption.

**Assumption 3.1.** (TFLs Assumption). Since $(x, y)$ is sampled randomly and uniformly from an unknown probability distribution with density $p(x, y)$, the conditional probability distribution of TFLs $\{p(y = i|\bar{y} = i, s = 1, x)\}_{i=1}^{K}$ is under the TFLs assumption as follows:

$$p(y = 1|\bar{y} = 1, s = 1, x) = p(y = 2|\bar{y} = 2, s = 1, x)$$
$$\vdots \tag{2}$$
$$= p(y = K|\bar{y} = K, s = 1, x)$$
$$= 1$$

It is worth noting that we cannot employ the conditional probability $p(y = i|x)$ in Eq.(1) directly since we do not have access to ordinary supervised data. Fortunately, we can use TFLs data to represent it by introducing the TFLs conditional probability $p(y = i, \bar{y} = j, s = 0|x)$.

**Lemma 3.2.** *Under the TFLs Assumption 3.1, the conditional probabilities $p(y = i|x)$ can be expressed as:*

$$p(y = i|x) = p(\bar{y} = i, s = 1|x) +$$
$$\sum_{j=1, j\neq i}^{K} p(y = i|\bar{y} = j, s = 0, x)p(\bar{y} = j, s = 0|x) \tag{3}$$

**Theorem 3.3.** *To deal with TFL learning problem, according to the Assumption 3.1 and Lemma 3.2, the classification risk $R(f)$ in Equation (1) could be rewritten as*

$$R_{TF}(f) = \mathbb{E}_{p(x,\bar{y},s=0)}\bar{\mathcal{L}}(f(x), \bar{y}) + \mathbb{E}_{p(x,\bar{y},s=1)}\mathcal{L}(f(x), \bar{y}) \tag{4}$$

*where $\bar{\mathcal{L}}(f(x), \bar{y}) = \sum_{i=1, i\neq j}^{K} p(y = i|\bar{y} = j, s = 0, x)\mathcal{L}(f(x), i)$. The proof is provided in the Appendix A.2.*

**Remark 3.4.** *To fully explore and leverage the prior knowledge of VLMs, we employ VLMs to precisely estimate conditional probability distributions $p(y = i|\bar{y} = j, s = 0, x)$ in Theorem 3.3. And then the empirical risk estimator can be expressed as:*

$$\hat{R}_{TF}(f) = \frac{1}{N_F} \sum_{l=1}^{N_F} \bar{\mathcal{L}}(f(x_l), \bar{y}_l) + \frac{1}{N_T} \sum_{l=1}^{N_T} \mathcal{L}(f(x_l), \bar{y}_l) \tag{5}$$

*where $N_F$ and $N_T$ denote the number of instances with binary TFL $s = 0$ and $s = 1$. Then, we can learn a multi-class classifier $f(x) : x \rightarrow \{1, \ldots, K\}$ by minimizing the proposed empirical approximation of the risk-consistent estimator in Eq (5).*

### 3.4. Multi-modal Prompt Retrieving

In this section, we introduce a convolutional-based **M**ulti-modal **P**rompt **R**etrieving (MPR) method to bridge the gap between the knowledge of VLMs and target learning tasks. MPR supplements discriminative features from weakly supervised data through cross-modal retrieval, reducing learning complexity while improving model robustness. Specifically, we retrieve visual and textual embeddings by learning a convolutional-based prompt network on top of CLIP. The learnable convolutional network retrieves domain-aware embeddings (e.g., culinary textures in Food-101), addressing CLIP's generic prompt limitations.

The overall architecture of the MPR is shown in Figure 2. Note that the base models of CLIP(Radford et al., 2021) is frozen in the entire training process. MPR is comprised of two distinct components, **T**extual **P**rompt **R**etrieving (TPR) and **V**isual **P**rompt **R**etrieving (VPR). They share one convolutional-based prompt network for prompt retrieving.

First of all, we select a matrix $\mathbf{M} \in \mathbb{R}^{H \times W \times B}$ whose elements are all initialized to 1. This matrix will be fed into a convolutional-based prompt network $g_{cnn}(\cdot)$ and the image encoder $g_I(\cdot)$ to obtain prompt embedding $q_p = g_I(g_{cnn}(\mathbf{M}))$.

For the TPR, we use the text prompt template "This is a photo of [CLS]"(Radford et al., 2021), where "[CLS]" represents category labels. By putting text prompts for all categories $\{P_i^T\}_{i=1}^{K}$ into the text encoder $g_T(\cdot)$, we obtain the text embeddings $\mathbb{Q}_T = \{q_i^T\}_{i=1}^{K}$ for all categories, where $q_i^T = g_T(P_i^T)$.

For the VPR, we randomly sample images $\{P_n^I\}_{n=1}^{C}$ from the dataset to create the retrieval image set, where $C$ denotes the image number of the retrieval image set. These images are then fed to the image encoder $g_I(\cdot)$ to obtain the image embeddings $\mathbb{Q}_I = \{q_n^I\}_{n=1}^{C}$, where $q_n^I = g_I(P_n^I)$.

Besides, we retrieve $K_T$ text embeddings and $K_I$ image embeddings that are most similar to the prompt embedding, respectively.

$$q^T = \left\{q_r^T\right\}_{r=1}^{K_T} = \underset{q_i^T \in \mathbb{Q}_T}{Top\text{-}K_T}(\cos(q_i^T, q_p))$$
$$q^I = \left\{q_r^I\right\}_{r=1}^{K_I} = \underset{q_n^I \in \mathbb{Q}_I}{Top\text{-}K_I}(\cos(q_n^I, q_p)) \tag{6}$$

where $\cos(\cdot, \cdot)$ denotes cosine similarity, and the $\underset{q_i \in \mathbb{Q}}{Top\text{-}K}(\cos(q_i, q_p))$ will retrieve top $K$ vectors with the highest similarity to vector $q_p$ from $\mathbb{Q}$. $K_T$ and $K_I$ are hyperparameters to balance TPR and VPR.

These embeddings are then flattened for further processing. Then we obtain the TPR embeddings $q_T$ and VPR

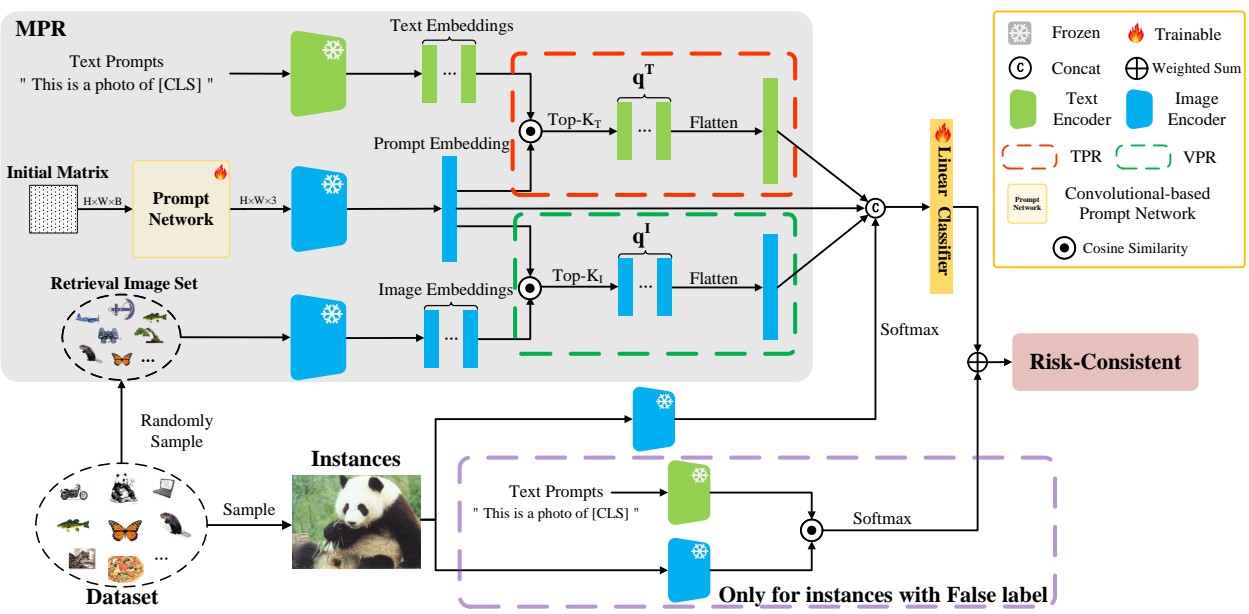

*Figure 2.* The architecture of TMP, including MPR and risk-consistent estimator. MPR retrieve visual and textual embeddings by learning a convolutional-based prompt network on top of CLIP. The goal of risk-consistent estimator is to explore and utilize the conditional probability distribution of TFLs.

embeddings $q_I$, which make up the MPR embeddings.

$$q_T = g_f(q_1^T, \cdots, q_{K_T}^T)$$
$$q_I = g_f(q_1^I, \cdots, q_{K_I}^I) \qquad (7)$$

The $g_f(\cdot)$ function flattens input vectors by reshaping them into a one-dimensional vector.

To the best of our knowledge, MPR is the first convolutional-based prompt learning approach for fine-tuning VLMs. Unlike directly optimizing the text or image itself (Zhou et al., 2022b; Bahng et al., 2022), Convolutional Neural Networks (CNNs) offer significant advantages in capturing local features. Direct image optimization, on the other hand, requires independent updates for each pixel, resulting in a lack of correlation between them. MPR enhances both textual and visual modalities by providing supplementary information without requiring additional data. Additionally, MPR is straightforward and relatively inexpensive in terms of computational resources compared to other multi-modal prompt learning methods.

### 3.5. Practical Implementation

In this section, we introduce the practical implementation of the proposed method.

**Conditional probability distribution.** Notice that minimizing $\hat{R}_{TF}$ requires estimating the conditional probability distributions $p(y = i|\bar{y} = j, s = 0, x)$ in Theorem 3.3. To fully explore and leverage the prior knowl-

edge of VLMs, we employ VLMs to precisely estimate $p(y = i|\bar{y} = j, s = 0, x)$. Specifically, we could get the conditional probability distributions of linear classifier $P_{LC}$, which is formulated as follows:

$$P_{LC} = Softmax(g_l(Concat(g_I(x), q_p, q_T, q_I))) \qquad (8)$$

where $g_l(\cdot)$ is a linear classifier and $Concat(\cdot)$ concatenates the given sequence of vectors. Besides, we obtain the conditional probability distributions $P_{CLIP}$ from CLIP, which can be formalized as follows:

$$P_{CLIP} = Softmax(cos(g_I(x), \mathbb{Q}_T)) \qquad (9)$$

Finally, we obtain the empirical conditional probability distribution through linear weighted sum method, which can be formalized as follows:

$$\hat{p}(y = i|\bar{y} = j, s = 0, x) = \lambda P_{LC} + (1 - \lambda)P_{CLIP} \qquad (10)$$

where $\lambda \in [0, 1]$ is a conditional probability hyperparameter that allows our model to simultaneously leverage the knowledge from VLMs and the learned model to enhance the performance of classification.

The conditional probability distribution $p(y = i|\bar{y} = j, s = 0, x)$ can be estimated as $\hat{p}(y = i|\bar{y} = j, s = 0, x)$. Then we can calculate the empirical risk-consistent cross-entropy loss based on the $\hat{p}(y = i|\bar{y} = j, s = 0, x)$, to optimize both the linear classifier and the convolutional-based prompt network.

**Model & Algorithm.** The convolutional-based prompt network comprises four convolutional layers (model architecture in Appendix A.3). For a comprehensive understanding, Algorithm 1 in Appendix A.4 outlines the overall procedure.

**Loss functions.** Many loss functions satisfy our method, such as logistic loss $\mathcal{L}(f(x), y) = \log(1 + e^{-yf(x)})$, MSE loss $\mathcal{L}(f(x), y) = (y - f(x))^2$, etc. In our experiments, we utilize the widely used cross-entropy loss function in multi-class classification $\mathcal{L}(f(x), y) = -y \log(f(x))$.

## 4. Experiments

### 4.1. Experimental Setup

**Dataset.** The efficacy of our method was evaluated on five distinct multi-class image classification datasets that feature both coarse-grained (CIFAR-100(Krizhevsky et al., 2009), Tiny ImageNet(Le & Yang, 2015) and Caltech-101(Fei-Fei et al., 2004)) and fine-grained (Food-101(Bossard et al., 2014) and Stanford Cars(Krause et al., 2013)) classification in different domains. For each dataset, the label of each image in the training set is replaced with the True-False Label (TFL), and the labels in the test set remain unchanged from the ground-truth labels. More information related to the datasets is shown in the Appendix A.5.

**Implementation details.** To ensure fair comparisons, for all experiments, we use CLIP with ViT-L/14 as the vision backbone, and employ the AdamW optimizer(Loshchilov & Hutter, 2019) for the linear classifier with an initial learning rate of $1e^{-3}$, a weight decay parameter set to 0.9, and the minimum learning rate of $5e^{-6}$. Unless otherwise noted, all models are trained for 50 epochs with a batch-size of 256 on a single NVIDIA RTX 4090 GPU. In our experiments, we employ the AdamW optimizer for the convolutional-based prompt network with an initial learning rate of $8e^{-2}$, a weight decay parameter set to 0.01, and the minimum learning rate of $5e^{-4}$. The hyperparameters $K_T$ and $K_I$ are set to 15 and 5, respectively. The size of matrix $\mathbf{M}$ is set to $224 \times 224 \times 1$.

**Compared methods.** To assess the efficacy of the proposed approach, a thorough evaluation is conducted through comparisons with weakly supervised learning methods, including semi-supervised learing (SSL) methods, partial-label learning (PLL) methods and complementary-label learning (CLL) methods. VLMs-based approaches are also considered. Specifically, the image encoders in the weakly supervised methods mentioned above were built using the same CLIP image encoder, transforming them into more competitive weakly supervised linear probe models. The key summary statistics for the compared methods are as follows:

- ORCA(Cao et al., 2022) and NACH(Guo et al., 2022): The SSL methods aiming to classify both seen and unseen classes effectively. In our experiments, we treat instances with $s = 1$ as supervised data and instances with $s = 0$ as unlabeled data.

- PaPi(Xia et al., 2023): An PLL method eliminating noisy positives and adopting a different disambiguation guidance direction. In our experiments, we treat instances with $s = 1$ as supervised data and instances with $s = 0$ as partial-labeled data. Then we consider all categories other than the randomly sampled label as the candidate label set, for partial-labeled data.

- CLL with WL(Gao & Zhang, 2021): A CLL method with a weighted loss. In our experiments, we treat instances with $s = 1$ as supervised data and instance with $s = 0$ as complementary-labeled data. Then we consider the randomly sampled label as the class label that the instance does not belong to for complementary-labeled data.

- CLIP Linear Probe(Radford et al., 2021) (CLIP LP): A VLMs-based approach which trains an additional linear classifier on top of CLIP's visual encoder. In our experiments, we employ three different settings: CLIP LPS, CLIP LPT, and CLIP LPP. CLIP LPS denotes the application of all labeled training data in a fully-supervised learning framework. CLIP LPT denotes learning using TFLs (i.e., only use instances with $s = 1$ as supervised data). CLIP LPP denotes the application of CLIP supervised labeled data in a fully-supervised learning framework.

To ensure that the only variable is the algorithm, we replace their original visual encoders with the same CLIP's visual encoder, and used the same linear classifier across all experiments.

### 4.2. Results of VLMs-based TFLs

In this section, we utilize the TFLs generated by CLIP with ViT-L/14. From the results presented in Table 2, it can be observed that the proposed method consistently outperforms all weakly supervised baselines by a large margin (over 10%), especially on fine-grained datasets such as Stanford Cars (over 60%). These results demonstrate that traditional weakly supervised learning methods struggle to effectively leverage the prior knowledge of VLMs. In contrast, our approach can more fully exploit the capabilities of VLMs.

Furthermore, our approach exhibits performance enhancements over other methods based on VLMs. Specifically, our method outperforms zero-shot CLIP on all datasets, with an average improvement of nearly 2%. Additionally, our method converges more rapidly than the CLIP linear probe.

*Table 2.* Comparison results on VLMs-based TFLs in terms of classification accuracy (the higher, the better). The best accuracy is highlighted in bold. TMP (VLMs) denotes the results on VLMs-based TFLs.

| | CIFAR-100 | Tiny ImageNet | Caltech-101 | Food-101 | Stanford Cars | Average |
|---|---|---|---|---|---|---|
| | VLMs Supervised Labels Learning | | | | | |
| CLIP LPP(Radford et al., 2021) | 77.4 | 74.71 | 88.74 | 93.30 | 71.56 | 81.14 |
| | Weakly Supervised Learning Methods | | | | | |
| ORCA(Cao et al., 2022) | 53.26 | 19.21 | 14.40 | 7.96 | 7.25 | 20.42 |
| NACH(Guo et al., 2022) | 64.42 | 35.09 | 21.39 | 12.62 | 4.53 | 27.61 |
| PaPi(Xia et al., 2023) | 63.73 | 41.50 | 43.27 | 81.94 | 10.19 | 48.13 |
| CLL with WL(Gao & Zhang, 2021) | 59.05 | 44.21 | 44.79 | 85.46 | 10.40 | 48.78 |
| | VLMs-based Methods | | | | | |
| Zero-shot CLIP(Radford et al., 2021) | 75.58 | 72.66 | 87.24 | 92.76 | 70.76 | 79.80 |
| CLIP LPT(Radford et al., 2021) | 25.25 | 18.24 | 22.63 | 66.69 | 4.96 | 27.55 |
| CLIP LPT (200 epochs)(Radford et al., 2021) | 27.86 | 20.26 | 25.07 | 69.35 | 5.53 | 29.61 |
| TMP (VLMs) | **78.22** | **75.14** | **89.14** | **93.52** | **72.52** | **81.71** |

*Table 3.* Comparison results on human-based TFLs in terms of classification accuracy (the higher, the better). The best accuracy is highlighted in bold. TMP (human) denotes the results on human-based TFLs data and TMP (VLMs) denotes the results on VLMs-based TFLs.

| | CIFAR-100 | Tiny ImageNet | Caltech-101 | Food-101 | Stanford Cars | Average |
|---|---|---|---|---|---|---|
| | Human Supervised Labels Learning | | | | | |
| CLIP LPS(Radford et al., 2021) | 85.81 | 85.31 | 96.76 | 94.94 | 87.71 | 90.11 |
| | Weakly Supervised Learning Methods | | | | | |
| ORCA(Cao et al., 2022) | 50.54 | 17.65 | 14.14 | 7.82 | 7.25 | 19.48 |
| NACH(Guo et al., 2022) | 63.46 | 31.78 | 15.09 | 8.51 | 5.57 | 24.89 |
| PaPi(Xia et al., 2023) | 60.69 | 40.80 | 47.06 | 80.31 | 6.00 | 46.97 |
| CLL with WL(Gao & Zhang, 2021) | 63.25 | 51.54 | 50.73 | 87.85 | 9.64 | 52.60 |
| | VLMs-based Methods | | | | | |
| Zero-shot CLIP(Radford et al., 2021) | 75.58 | 72.66 | 87.24 | 92.76 | 70.76 | 79.80 |
| CLIP LPT(Radford et al., 2021) | 25.60 | 21.37 | 21.36 | 68.23 | 3.69 | 28.05 |
| CLIP LPT (200 epochs)(Radford et al., 2021) | 28.06 | 23.39 | 24.31 | 70.65 | 4.24 | 30.12 |
| TMP (VLMs) | 78.22 | 75.14 | 89.14 | 93.52 | 72.52 | 81.71 |
| TMP (human) | **78.72** | **75.84** | **90.60** | **93.55** | **72.60** | **82.07** |

Surpassing the performance of the CLIP linear probe trained for 200 epochs, we achieve better results after just 50 epochs. This is due to the limited number of instances with $s = 1$ that the CLIP linear probe can utilize. Besides, the results of TMP on all five datasets are better than VLMs supervised learning results. TMP uses less supervisory information and achieves better results, which demonstrates the potential of TFL as a labeling setting.

### 4.3. Results of Human-based TFLs

We use the ground-truth labels to generate TFLs for the training set. All remaining settings are identical to those in section 4.2. Table 3 exhibits a similar trend to Table 2. Compared to other methods, our method achieves the best results on all datasets, which substantiates the effectiveness of TMP. Besides, our method achieves performance compa-

rable to fully supervised approaches on Food-101. These experimental results demonstrate that our method effectively bridge the gap between knowledge of CLIP and target learning tasks. It is worth noting that in some weakly supervised methods, training with TFLs generated by CLIP can achieve even better results compared to those presented in Table 3. Specifically, the performance of the CLL method on CIFAR-100 has improved by over 4%. A heuristic reason for this is that TFLs generated by CLIP may correct inherent noise in the original dataset.

**VLMs-based vs. Human-based TFLs.** As shown in the last two rows of Table 3, there is no significant performance difference between TMP (human) and TMP (VLMs) (with an average difference of no more than 0.3%), indicating that VLMs are capable of generating sufficiently high-quality TFLs.

*Table 4.* Experimental results on the influence of MPR. w/o denotes without the component. Experiments are performed on human-based TFLs data.

|  | Caltech-101 | Food-101 | Stanford Cars | Average |
| --- | --- | --- | --- | --- |
| TMP(human) | 90.60 | 93.55 | 72.60 | 85.58 |
| w/o MPR | 88.81 | 93.47 | 71.25 | 84.51 |
| w/o TPR | 89.80 | 93.53 | 72.25 | 85.19 |
| w/o VPR | 90.23 | 93.52 | 72.53 | 85.43 |

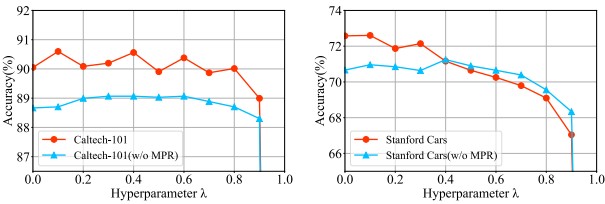

*Figure 3.* Experimental results on the influence of the conditional probability hyperparameter $\lambda$. Experiments are performed on human-based TFLs data.

### 4.4. Influence of MPR

In Table 4, we explore the effectiveness of our proposed MPR method, which consists of TPR and VPR components, on a coarse-grained dataset (Caltech-101) and two fine-grained datasets (Food-101 and Stanford Cars). We conducted experiments by individually removing the TPR and VPR, as well as removing the MPR as a whole. The results show that the removal of each component leads to some degree of performance degradation. Specifically, each component (TPR and VPR) leads to an average performance improvement of more than 0.5%. This improvement confirms the discussion in earlier section that MPR provides more related information to bridge the gap between the knowledge of VLMs and target learning tasks.

### 4.5. Influence of the Hyperparameter $\lambda$

We check how performance varies w.r.t. $\lambda$ on a coarse-grained dataset (Caltech-101) and a fine-grained dataset (Stanford Cars). Figure 3 shows that as $\lambda$ increases from 0 to 0.9, there is a gradual decline in performance. Specifically, these two datasets achieved the best accuracy at $\lambda = 0.1$ (i.e., 90.60 on Caltech-101 and 72.60 on Stanford Cars), which demonstrates the necessity of simultaneously leveraging the knowledge from VLMs and the learned model. Note that our method demonstrates stable performance when the conditional probability hyperparameter $\lambda$ is within the range $[0, 0.9]$. The reason may be that TMP could adaptively explore the relationship between VLMs and the learned model to effectively estimate the probability distribution.

### 4.6. Comparison of Training Cost

In this section, we compare the training cost of our proposed TMP method with those of various weakly supervised learn-

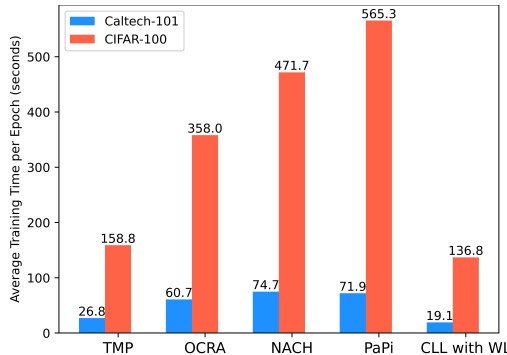

*Figure 4.* Comparison of training cost between TMP and weakly supervised learning methods. The numbers in the figure represent the average time (in seconds) required to train each method for single epoch. The experiments utilize the CIFAR-100 and Caltech-101 datasets, conducted on a single NVIDIA RTX 4090 GPU.

ing methods. The experiments utilize the CIFAR-100 and Caltech-101 datasets, conducted on a single NVIDIA RTX 4090 GPU, with all other settings consistent with those of the previous experiments. As shown in Figure 5, we measure the time (in seconds) required to train each method for one epoch. TMP significantly reduces training time compared to weakly supervised learning methods. Specifically, the training time for ORCA, NACH, and PaPi exceeds twice that of TMP. Additionally, the training times for TMP and CLL with WL are comparable. This advantage is attributed to the fact that weakly supervised methods often necessitate multiple iterations over unlabeled data within a single epoch. Our experiments demonstrate that TMP achieves higher accuracy while requiring less training time.

## 5. Conclusion

In this paper, we investigate a novel weakly supervised learning problem called learning from True-False Labels (TFLs), which can significantly enhance the quality and efficiency of annotation. In this novel labeling setting, the TFL indicates whether an instance belongs to the label that is randomly and uniformly sampled from the candidate label set. We theoretically derive a risk-consistent estimator to explore and utilize the conditional probability distribution information of TFLs. Besides, we introduce a novel prompt learning method called MRP learning, which can bridge the gap between the knowledge of VLMs and target learning tasks. The experimental results demonstrate the effectiveness of the proposed TFL setting and MRP learning method.

**Limitations and future directions.** A limitation of this method is that it does not use test data to improve model performance. In the future, it is interesting to explore a test-time prompt retrieving method which can learn adaptive prompts with a single test instance.

## Acknowledgements

This work was supported by the National Natural Science Foundation of China (No.62306320, 61976217), the Open Project Program of State Key Lab. for Novel Software Technology (No. KFKT2024B32), and the Natural Science Foundation of Jiangsu Province (No. BK20231063).

## Impact Statement

The primary objective of this paper is to advance the field of machine learning. While our work carries numerous potential societal implications, we deem it unnecessary to explicitly highlight them within this context.

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

# A. Appendix / supplemental material

## A.1. Proof of Lamme 3.2

**Lemma A.1.** *Under the TFLs Assumption 3.1, the conditional probabilities $p(y = i|x)$ can be expressed as:*

$$p(y = i|x) = p(\bar{y} = i, s = 1|x) + \sum_{j=1, j \neq i}^{K} p(y = i|\bar{y} = j, s = 0, x)p(\bar{y} = j, s = 0|x) \tag{11}$$

*Proof.* According to Assumption 3.1, Bayes Rule and Total Probability Theorem,

$$
\begin{aligned}
p(y = i|x) =& p(y = i, s = 1|x) + p(y = i, s = 0|x) \\
=& \sum_{j=1}^{K} p(y = i, \bar{y} = j, s = 1|x) + \sum_{j=1, j \neq i}^{K} p(y = i, \bar{y} = j, s = 0|x) \\
=& \sum_{j=1}^{K} p(y = i|\bar{y} = j, s = 1, x)p(\bar{y} = j, s = 1, x) + \sum_{j=1, j \neq i}^{K} p(y = i, \bar{y} = j|s = 0, x)p(s = 0|x) \\
=& p(y = i|\bar{y} = i, s = 1, x)p(\bar{y} = i, s = 1|x) + \sum_{j=1, j \neq i}^{K} p(y = i|\bar{y} = j, s = 0, x)p(\bar{y} = j|s = 0, x)p(s = 0|x) \\
=& p(\bar{y} = i, s = 1|x) + \sum_{j=1, j \neq i}^{K} p(y = i|\bar{y} = j, s = 0, x)p(\bar{y} = j, s = 0|x).
\end{aligned}
$$

$\square$

## A.2. Proof of Theorem 3.3

**Theorem A.2.** *To deal with TF label learning problem, according to the Assumption 3.1 and Lemma 3.2, the classification risk $R(f)$ in Equation (1) could be rewritten as*

$$R_{TF}(f) = \mathbb{E}_{p(x,\bar{y},s=0)} \bar{\mathcal{L}}(f(x), \bar{y}) + \mathbb{E}_{p(x,\bar{y},s=1)} \mathcal{L}(f(x), \bar{y}) \tag{12}$$

*where $\bar{\mathcal{L}}(f(x), \bar{y}) = \sum_{i=1, i \neq j}^{K} p(y = i|\bar{y} = j, s = 0, x)\mathcal{L}(f(x), i)$.*

*Proof.* According to the Assumption 3.1 and Lemma 3.2

$$
\begin{aligned}
R_{TF}(f) =& \mathbb{E}_{p(x,y)}[\mathcal{L}(f(x), y)] \\
=& \mathbb{E}_{x \sim \mu} \sum_{i=1}^{K} p(y = i|x)\mathcal{L}(f(x), i) \\
=& \mathbb{E}_{x \sim \mu} \sum_{i=1}^{K} \sum_{j=1, j \neq i}^{K} p(y = i|\bar{y} = j, s = 0, x)p(\bar{y} = j, s = 0|x)\mathcal{L}(f(x), i) + \mathbb{E}_{x \sim \mu} \sum_{i=1}^{K} p(\bar{y} = i, s = 1|x)\mathcal{L}(f(x), i) \\
=& \mathbb{E}_{x \sim \mu} \sum_{j=1}^{K} p(\bar{y} = j, s = 0|x) \sum_{i=1, i \neq j}^{K} p(y = i|\bar{y} = j, s = 0, x)\mathcal{L}(f(x), i) + \mathbb{E}_{x \sim \mu} \sum_{i=1}^{K} p(\bar{y} = i, s = 1|x)\mathcal{L}(f(x), i) \\
=& \mathbb{E}_{p(x,\bar{y},s=0)} \sum_{i=1, i \neq j}^{K} p(y = i|\bar{y} = j, s = 0, x)\mathcal{L}(f(x), i) + \mathbb{E}_{p(x,\bar{y},s=1)} \mathcal{L}(f(x), \bar{y}) \\
=& \mathbb{E}_{p(x,\bar{y},s=0)} \bar{\mathcal{L}}(f(x), \bar{y}) + \mathbb{E}_{p(x,\bar{y},s=1)} \mathcal{L}(f(x), \bar{y}).
\end{aligned}
$$

$\square$

### A.3. The specific architecture of the convolutional-based prompt network

As shown in Figure 5, the convolutional-based prompt network consists of four CNN blocks, each with varying input and output channel configurations. Within each CNN block, there is a convolutional layer with a kernel size of 3 and a stride of 1, followed by a batch normalization layer, a Leaky ReLU activation layer, and a dropout layer.

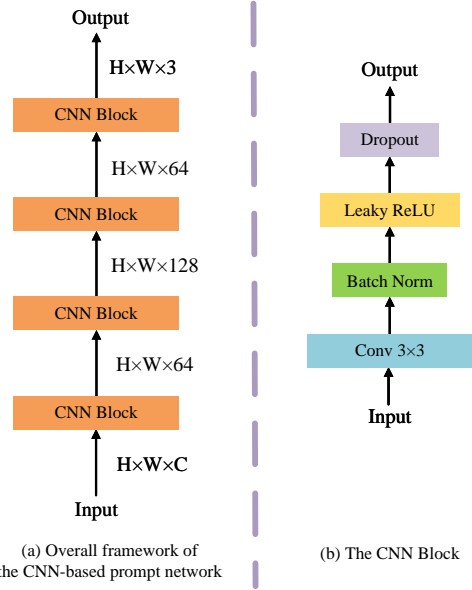

(a) Overall framework of the CNN-based prompt network

(b) The CNN Block

Figure 5. Overall framework of the convolutional-based prompt network

The CNN-based network used in our study contains 150,531 parameters, which is comparable to the 150,528 pixels in an image of size $224 \times 224 \times 3$. Despite similar numbers of parameters, CNN offer significant advantages in capturing local features, unlike direct image optimization, which would require independent updates for each pixel, leading to a lack of correlation between pixels and computational inefficiency. Additionally, CNN offer key advantages like reduced computational complexity through parameter sharing and parallel execution on hardware accelerators.

### A.4. Overall algorithm procedure

Algorithm 1 illustrates the overall algorithm procedure. Through this process, we can learn a high-quality linear classifier and a convolutional-based prompt network. This convolutional-based prompt network retrieve multi-modal prompts to bridge the gap between knowledge of VLMs and target learning tasks. To ensure stable optimization of the parameters in the convolutional-based prompt network, we introduce a hyperparameter $m$ to control the process.

### A.5. The details of datasets

In this section, we provide a detailed description of datasets used in our experiments.

- CIFAR-100(Krizhevsky et al., 2009): A coarse-grained dataset comprising 60,000 color images divided into 100 classes. Each image is given in a $32 \times 32 \times 3$ format, and each class contains 500 training images and 100 test images.

- Tiny-ImageNet(Le & Yang, 2015): A coarse-grained dataset consists of 100,000 color images divided into 200 classes. Each image is given in a $64 \times 64 \times 3$ format, and each class contains 500 training images, 50 validation images and 50 test images.

- Caltech-101(Fei-Fei et al., 2004): A coarse-grained dataset comprises images from 101 object categories and a background category that contains the images not from the 101 object categories. Each object category contains approximately 40 to 800 images, with most classes having about 50 images. The image resolution is approximately 300×200 pixels.

---

**Algorithm 1** TFL learning via MPR

---

**Input:** The TF labeled training set $D_{TF} = \{(x_i, (\bar{y}_i, s_i))\}_{i=1}^{N}$; The convolutional-based prompt network $g_{cnn}(\cdot)$; A matrix $\mathbf{M}$, whose elements are all 1; The CLIP's image encoder $g_I(\cdot)$; The number of epochs $T$; The Stability Optimization hyperparameter $m$;

**Output:** Model parameter $\theta_1$ for the linear classifier; Model parameter $\theta_2$ for $g_{cnn}(\cdot)$

1: **for** $t = 0$ to $T$ **do**
2:     Shuffle $D_{TF} = \{(x_i, (\bar{y}_i, s_i))\}_{i=1}^{N}$ into $B$ mini-batches;
3:     $v_p = g_I(g_{cnn}(\mathbf{M}))$;
4:     Calculate $v_T$ and $v_I$ by Eq.(7);
5:     **for** $b = 0$ to $B$ **do**
6:         Fetch mini-batch $D_B$ from $D_{TF}$;
7:         Calculate $\hat{p}(y = i|\bar{y} = j, s = 0, x)$ by Eq.(10);
8:         Update the linear classifier's parameters $\theta_1$ by $\hat{R}_{TF}$ in Eq.(5);
9:         **if** $t\%m = 0$ **then**
10:            Update the convolutional-based prompt network's parameters $\theta_2$ by $\hat{R}_{TF}$ in Eq.(5);
11:         **end if**
12:     **end for**
13: **end for**

---

- Food-101(Bossard et al., 2014): A fine-grained dataset in the food domain, comprising 101,000 images divided into 101 food categories. Each class contains 750 training images and 750 test images. The labels for the test images have been manually cleaned, while the training set contains some noise.

- Stanford Cars(Krause et al., 2013): A fine-grained dataset in the car domain, comprising 16,185 images categorized into 196 car classes. The data is divided into almost a 50-50 train/test split with 8,144 training images and 8,041 testing images. Categories are typically at the level of Make, Model, Year. The images are 360×240 pixels.

