# OpenReview forum: "Learning from True-False Labels via Multi-modal Prompt Retrieving"
_ICML.cc/2025/Conference — ICML 2025 poster_

### Official Review · Reviewer_peko · 2025-02-19

**Overall Recommendation:** 3

**Summary:**

This paper propose a novel weakly supervised labeling setting, namely True-False Labels (TFLs) which can achieve high accuracy when generated by pre-trained Vision-Language Models (VLM). Moreover, the paper derived a risk-consistent loss for this setting and propose a convolutional-based Multi-modal Prompt Retrieving (MRP) method to bridge the gap between the knowledge of VLMs and target learning tasks.  Experimental results demonstrate the effectiveness of the proposed TFL setting and MRP learning method.

**Claims And Evidence:**

Yes.

**Essential References Not Discussed:**

No.

**Experimental Designs Or Analyses:**

Yes, I have carefully reviewed the experimental designs and analyses, and several issues need to be addressed to ensure the soundness and validity of the results:
1. **Unusual Weak Supervision Results**: The results of the weakly supervised learning methods in Table 2 and Table 3 appear to be quite unusual. Specifically, the fine-tuned results are significantly worse than the zero-shot results, which is counterintuitive and raises concerns about the validity of the experimental setup or implementation. The authors should provide more detailed training configurations and hyper parameters for these methods to ensure reproducibility. Additionally, they should carefully examine whether the poor performance is caused by suboptimal training details, such as learning rates, optimization strategies, or data pre-processing. Without a thorough investigation and clarification, the credibility of the reported results remains questionable.
2. **Insufficient Baseline Comparisons**: The current experimental design lacks comprehensive comparisons with relevant baselines, which limits the ability to assess the effectiveness of the proposed method. Specifically:
   - For only using supervised samples, some state-of-the-art few-shot fine-tuning methods such as [1] should be included as baselines.
   - For the semi-supervised setting, advanced semi-supervised methods specifically designed for fine-tuning models [2, 3] should be compared.
   - For complementary labeling, more sophisticated methods utilizing complementary labels such as [4] should be incorporated
   - Finally, self-training methods under other weak supervision settings [5] should also be added to the comparison.

By expanding the comparison to these relevant baselines, the authors can provide a more comprehensive evaluation of their method's performance and better highlight its potential advantages over existing approaches. Without such comparisons, it is difficult to assess whether the proposed method truly advances the state-of-the-art for VLMs fine-tuning.

In summary, while the paper explores an interesting direction, the experimental design and analysis need significant improvements to ensure the validity and soundness of the results. Addressing these issues would strengthen the paper's contributions and provide a more convincing evaluation of the proposed method.

Reference:

[1] Zhang, RenRui et al. Tip-Adapter: Training-free CLIP-Adapter for Better Vision-Language Modeling

[2] Gan, Kai ea al. Erasing the Bias: Fine-Tuning Foundation Models for Semi-Supervised Learning

[3] Wang, XuDong et al. Debiased learning from naturally imbalanced pseudo-labels

[4] Wang, Wei et al. Learning with Complementary Labels Revisited: The Selected-Completely-at-Random Setting Is More Practical

[5] Zhang, Jiahan et al. Candidate Pseudo label Learning: Enhancing Vision-Language Models by Prompt Tuning with Unlabeled Data

**Methods And Evaluation Criteria:**

The proposed methods have some limitations that reduce the significance of the paper's contributions:

Firstly, the proposed weak supervision setting appears to be a straightforward combination of full supervision and complementary labeling. The authors do not sufficiently justify why and how this particular weak supervision setting is advantageous for VLM fine-tuning compared to existing weak supervision setting. For instance, prior work such as [1] using partial label setting for fine-tune VLM. The paper would benefit from a more thorough comparison and analysis of how the proposed approach improves upon or differs from existing weak supervision setting in the context of VLM fine-tuning.

The proposed  methods lacks novelty, as it is essentially equivalent to the widely-used self-training approach in weak supervision learning. Specifically, the proposed method is essentially equivalent to a two-stage self-training approach. In the first stage, complementary labels or ground-truth labels are assigned to the samples. In the second stage, the model learns from the samples with complementary labels by utilizing the predicted posterior probabilities. This formulation aligns closely with conventional self-training frameworks, where pseudo-labels are iteratively generated and refined to improve model performance. The authors should clarify how their approach differs from or advances beyond this well-established paradigm, as the current presentation does not sufficiently highlight novel methodological contributions.

In summary, while the paper explores an interesting direction, the lack of significant novelty in the proposed methods and insufficient comparison to existing approaches reduce the overall impact of the work. The authors should address these limitations by providing a more thorough analysis of how their approach advances the state-of-the-art real-world VLMs fine tune.

[1] Zhang, Jiahan et al. Candidate Pseudolabel Learning: Enhancing Vision-Language Models by Prompt Tuning with Unlabeled Data. ICML (2024).

**Other Comments Or Suggestions:**

I don't have any further comments of suggestions.

**Other Strengths And Weaknesses:**

Other Strengths:
1. **Novel Weak Supervision Annotation Scheme**: The introduction of True-False labels as a new weak supervision annotation method is a creative and potentially impactful contribution. This approach leverages the capabilities of Vision-Language Models (VLMs) to automate the annotation process, offering a scalable and efficient alternative to traditional weak supervision techniques.
2. **Competitive Performance**: The proposed fine-tuning method, tailored to the True-False labels, demonstrates competitive performance on the evaluated benchmarks. This suggests that the method is effective in leveraging weak supervision signals to improve model performance.
3. **Relevance to Broader Research Trends**: The work aligns with the growing interest in reducing reliance on expensive manual annotations while maintaining or improving model performance. The proposed method could inspire further research into automated weak supervision strategies for fine-tuning foundation models.

**Questions For Authors:**

**Question 1: Comparison with Standard Self-Training and Prior Work**

The proposed method appears to be essentially equivalent to a two-stage self-training approach:
1. In the first stage, complementary labels or ground-truth labels are assigned to the samples.
2. In the second stage, the model learns from the samples with complementary labels by utilizing the predicted posterior probabilities.

While this formulation shares similarities with standard self-training and prior work such as [1], it is unclear why the proposed method is superior. Specifically:
- **Compared to standard self-training**: What are the advantages of using complementary labels (True-False labels) over traditional pseudo-labels? Is there a theoretical or empirical justification for why this approach leads to better performance or faster convergence?
- **Compared to [1]**: How does the proposed method improve upon or differ from the weak supervision strategies explored in [1]? For instance, does the use of True-False labels provide better robustnessor more efficient utilization of unlabeled data?

I hope authors provide a more in-depth discussion, supported by either theoretical analysis, empirical results, or intuitive insights, to clarify the unique benefits of their approach over existing methods.

**Question 2: Poor Performance of Previous Weak Supervision Methods**

The results of previous weak supervision methods, as shown in Table 2 and Table 3, are notably poor—even worse than the zero-shot performance. This raises significant concerns and requires further explanation:
1. **Potential Causes**: Could the poor performance be attributed to suboptimal training configurations, such as inappropriate learning rates, insufficient training epochs, or inadequate hyper parameter tuning? Alternatively, is there an inherent limitation in the design of these weak supervision methods that makes them unsuitable for the task or dataset at hand?
2. **Proposed Method's Advantages**: The authors should clearly explain how their training method differs from previous weak supervision approaches and why it avoids the pitfalls that led to the poor performance of those methods.

Reference:

[1] Zhang, Jiahan et al. Candidate Pseudolabel Learning: Enhancing Vision-Language Models by Prompt Tuning with Unlabeled Data. ICML (2024).

I appreciate the authors' efforts in this work. I would like to raise my score if my concerns were addressed.

**Relation To Broader Scientific Literature:**

The key contributions of this paper are related to the broader scientific literature in the following ways:

1.**Novel Weak Supervision Annotation (True-False Labels)**: Compared to prior work, such as [1], this paper proposes a new weak  supervision annotation scheme—True-False labels—specifically designed for fine-tuning foundation models. This annotation can be automatically generated by VLMs, which distinguishes it from traditional weak supervision approaches that often rely on manual or heuristic labeling. This contribution addresses a gap in the literature by providing a more scalable and efficient way to generate weak supervision signals for fine-tuning.
2. **Fine-Tuning Method for True-False Labels**: Building on the proposed annotation scheme, the authors design a fine-tuning method tailored to True-False labels, which demonstrates competitive performance. This methodological advancement extends the existing literature on weak supervision by showing how such annotations can be effectively utilized to improve model performance. The results suggest that True-False labels, despite their simplicity, can serve as a viable alternative to more complex weak supervision strategies.

Reference:

[1]  Zhang, Jiahan et al. Candidate Pseudo label Learning: Enhancing Vision-Language Models by Prompt Tuning with Unlabeled Data. ICML (2024).

**Theoretical Claims:**

Yes.  All proofs for theoretical claims in this  paper are correct.

---

> ### Author Rebuttal · Authors · 2025-04-01
>
> Thank you for your detailed and constructive comments. We will address each concern point by point:
> **Q1&Method (Novelty & Comparison with Self-Training)**
> R1: Our TFL framework introduces three key innovations that fundamentally address limitations in standard self-training and prior work like Candidate Pseudo-Label Learning[1]:
> 1. **Random Label Sampling as Implicit Regularization**:
> TFL employs uniform random label sampling as implicit regularization to prevent model overfitting. This stochastic mechanism in TFL, analogous to established techniques like dropout, SGD and random data augmentation, promotes better generalization solutions [3]. Unlike confidence-based pseudo-labeling in self-training and [1], which amplifies CLIP's inherent biases through error propagation [1,2].
> 2. **High-Accuracy TFL for Bias Amplification Mitigation**
> Prior studies [2,3] have demonstrated that pseudo-labeling noise can create a cumulative effect on classes with inaccurate pseudo-labels, amplifying the model's inherent bias toward certain classes. The experimental results in Table 6 of [1] further support this conclusion, showing that accuracy improvements tend to be more pronounced on datasets where Zero-shot CLIP initially performs better. This observation aligns with TFL’s design motivation to mitigate bias amplification through stochastic sampling. TFL achieves over 99% annotation accuracy compared to the 85% accuracy shown in [1] (Fig.2). This significant improvement in labeling precision (via confidence-ranking-independent label generation) substantially reduces noise propagation.
> 3. **Hybrid Supervision Mechanism**
> TFL integrates strong supervision from retained true labels to provide semantic correction anchors, whereas [1] relies solely on candidate pseudo-label. The utilization of hybrid supervision is mathematically formalized through our risk-consistent estimator (Eq.5). This hybrid labels provide explicit optimization direction, enhances noise robustness and improves stability in complex scenarios.
>
> **Q2&E1 (Poor Performance of Weak Supervision Baselines)**
> R2: We rigorously verified our implementation (code available at [Anonymous GitHub](https://anonymous.4open.science/r/TMP-2D10)) under identical configurations (epochs, architecture, hyperparameters). The observed performance gap primarily arises from **Large Label Space Challenges**:
> - **TFL data contains numerous unseen class for semi-supervised learning** (e.g., 196 classes with only 42 supervised samples in Stanford Cars). This results in most classes lacking supervised samples. Existing semi-supervised methods for unknown classes fail to handle such a high proportion of unseen categories, leading to catastrophic performance collapse.
> - **The candidate sets generated by TFL are excessively large for partial/complementary-label learning**. Oversized candidate sets (>100 classes) severely degrade disambiguation capabilities. In the weak supervision community, Complementary-label learning[5] and partial-label learning[6] typically uses candidate sets containing ≤30 classes.
> Our method overcomes this through:
>    - MPR(Sec. 3.4)
>    - CLIP prior integration(Eq. 9-10)
>
> **E2&Method (Baseline Comparisons)**
> R3: In our experiments, we have conducted comprehensive comparisons with relevant works (Tables 2&3), consistently demonstrating superior performance. We further expanded comparisons to include: Few-shot methods(Tip-Adapter), Semi-supervised VLM fine-tuning approaches(FineSSL), More sophisticated methods utilizing complementary labels(DebiasPL), Advanced complementary-label methods(SCARCE). As shown below:
> | Method | CIFAR-100 | Caltech-101 |
> |:---:|:---:|:---:|
> | Tip-Adapter  | 76.47 | 86.77 |
> | Tip-Adapter-F | 77.40 | 86.81 |
> | FineSSL | 67.28 | 28.72 |
> | DebiasPL | - | 63.40 |
> | SCARCE | 44.06 | 39.39 |
> | CPL+LaFTer | 77.30 | 93.40 |
> | TMP | 78.72 | 90.60 |
>
> Notably, TMP achieves comparable performance to CPL+LaFTer on CIFAR-100 and Caltech-101, despite the fact that CPL+LaFTer requires more resources, such as leveraging additional LLM knowledge, and utilizes iterative pseudo-label refinement (T=10). These results conclusively validate TMP's effectiveness through its noise-robust probability estimation framework.
> Reference:
> [1]	Zhang J, et al. Candidate Pseudolabel Learning: Enhancing Vision-Language Models by Prompt Tuning with Unlabeled Data, ICML, 2024.
> [2]	Menghini C, et al.. Enhancing clip with clip: Exploring pseudolabeling for limited-label prompt tuning, NeurIPS, 2023.
> [3]	Wang X., et al. Debiased Learning from Naturally Imbalanced Pseudo-Labels, CVPR, 2022.
> [4]	Ali A, et al. The implicit regularization of stochastic gradient flow for least squares, ICML, 2020.
> [5]	Wang, Wei et al. Learning with Complementary Labels Revisited: The Selected-Completely-at-Random Setting Is More Practical, ICML 2024.
> [6]	Xia S, et al. Towards effective visual representations for partial-label learning, CVPR 2023.

---

> > ### Comment · Reviewer_peko · 2025-04-02
> >
> > Since most of my concerns have been resolved, I am inclined to recommend acceptance of the manuscript now.

---

### Official Review · Reviewer_X5ci · 2025-03-09

**Overall Recommendation:** 3

**Summary:**

This paper proposes a novel weakly supervised setting called True-False Labels (TFLs), leveraging VLM to reduce the difficulty of manual annotation. TFLs indicates whether a sample belongs to a label randomly and uniformly sampled from a candidate label set. In addition, this paper derives a risk-consistent estimator to explore and utilize the conditional probability distribution information of TFLs and introduces a Multimodal Prompt Retrieval to bridge the gap between VLM knowledge and the target learning task.

**Claims And Evidence:**

The paper presents a relevant and important problem, and the claims made are generally supported by evidence. However, the motivation behind the proposed framework is not sufficiently clear, making it difficult to fully understand how the proposed approach can effectively address the problem. The lack of clarity regarding the framework's motivation undermines the convincingness of the presented evidence and solution.

**Essential References Not Discussed:**

I have not identified any essential related works that are missing from the paper at this time.

**Experimental Designs Or Analyses:**

The analysis of the hyperparameter $\lambda$ lacks convincing evidence, as the performance improves when its value is small. This raises concerns about the lack of support for the effectiveness of TMP.

**Methods And Evaluation Criteria:**

Yes, the proposed methods and evaluation criteria are suitable for the problem and application at hand.

**Other Comments Or Suggestions:**

Please refer to the Questions For Authors.

**Other Strengths And Weaknesses:**

**Strengths:**
1. This paper proposes a novel weakly supervised learning annotation method, expanding the scope of weak supervision approaches.
2. The integration of VLMs to address weak supervision is an effective combination and is a promising research direction.

**Weaknesses:**
1. The introduction spends a significant portion explaining label annotation, which makes the latter part of the introduction insufficient and difficult to understand the motivation. Additionally, there is a lack of smooth transition to the methodology, making the proposed method less understandable.
2. The overall organization of the paper could be improved in terms of readability. In addition, Table 1 lacks necessary explanations, making it difficult to understand the meaning of certain columns. Improving readability would enhance clarity.

**Questions For Authors:**

1. Assumption 1 appears problematic, as it seems to disregard cases where TFL are incorrect.
2. Why was prompt retrieving designed? The method lacks sufficient explanation, and I did not fully understand the motivation behind introducing the TMP framework. Could this be clarified by providing further details in the introduction and starting from Section 3.4?
3. In Equation (9), does $x$ not need to go through the image encoder?
4. When the hyperparameter $\lambda$ is small, the performance improves, but this seems to lack evidence supporting the effectiveness of TMP.
5. In right-hand side of line 372, the paper states that "our approach has achieved results approaching those of the fully supervised method". Does this refer to the first row of the table 3? If yes, there is still a noticeable gap, so it may be worth reconsidering whether this statement is appropriate.
6. In line 718 of the appendix, how is the hyperparameter $m$ determined? This is not explained in the paper.

**Relation To Broader Scientific Literature:**

The paper builds upon two key areas of research: weakly supervised learning and vision-language models (VLMs). It primarily leverages the strong generalization ability of VLMs to address weak supervision and proposes a novel annotation strategy. In terms of problem formulation, the work is more closely related to prior research on pseudo-label learning.

**Theoretical Claims:**

Yes. Assumption 1 does not account for the case where the TFL is incorrect. Other parts of the proofs seem to be correct with no issues.

---

> ### Author Rebuttal · Authors · 2025-04-01
>
> Thank you for your detailed and constructive comments. We will address each concern point by point:
> **Q1: Assumption 1 appears problematic**
> R1: In fact, Assumption 1 emphasizes that consistent classifiers can learn from pre-existing VLM-generated or human-annotated TFLs data, i.e., TFLs consistent learn assumption, rather than focus on the results of labeling the data must be perfect.
> Furthermore, high-performance VLMs exhibit minimal annotation errors, with accuracy levels comparable to or surpassing human performance [1,2]. This suggests that they can satisfy the TFLs consistent learn assumption. Besides, in real-world scenarios, the goal of TFLs labeling is to mitigate noise introduced by VLMs-based annotations, ensuring that the labeled data adheres to the TFLs consistent learn assumption as closely as possible.
>
> **Q2&W1: Motivation for MPR requires clarification**
> R2: We have described the motivation for MPR in lines 104-109 and 290-294, among others, in the paper. More details are as follows:
> 1) **Weak Supervision Enhancement**: MPR supplements discriminative features from weakly supervised data through cross-modal retrieval, reducing learning complexity while improving model robustness.
> 2) **Task-Specific Adaptation** (Section 3.4): Our learnable convolutional network retrieves domain-aware embeddings (e.g., culinary textures in Food-101), addressing CLIP's generic prompt limitations.
> 3) **Modality Alignment** (Eq.7-9): By dynamically aggregating Top-K visual and textual prompts, MPR enhances modality consistency while preserving VLM knowledge.
> Ablation studies (Table 4) confirm MPR contributes 1.07% average accuracy improvement across datasets. We will strengthen the methodological motivation in Section 1.
>
> **Q3: Equation (9) image encoder clarification**
> R3: The revised formulation is:
> $P_{CLIP} = \text{Softmax}(\cos(g_I(x), \mathbb{Q}_T))$
> where $g_I(\cdot)$ denotes CLIP's image encoder. We will update Eq.9 and ensure consistency in all algorithm descriptions.
>
> **Q4: Effectiveness of TMP**
> R4: As shown in Tables 2&3, TMP achieves optimal performance across all datasets. Notably, optimal performance never occurs at λ=0. This demonstrates that our balanced integration strategy (λ=0.1) optimally combines VLM knowledge with task-specific adaptation, thereby validating TMP's effectiveness.
>
> **Q5: Line 372's "approaching fully supervised" claim may be overstated**
> R5: Compared to baseline methods, our approach achieves performance approaching fully supervised levels on specific datasets. For example, on the Food-101 dataset (93.55% vs. 94.94% fully supervised). However, a 15% performance gap persists on the Stanford Cars dataset. We have revised the statement to: "Our method achieves performance comparable to fully supervised approaches on Food-101."
>
> **Q6: Hyperparameter m determination**
> R6: These hyperparameter $m$ were selected through experimentation to strike a tradeoff between precision and efficiency (when m=1, the accuracy is the highest, but the training cost is relatively high).
> Reference:
> [1] Street W, et al. LLMs achieve adult human performance on higher-order theory of mind tasks, arXiv, 2024.
> [2] Kapania S, et al. "Because AI is 100% right and safe": User attitudes and sources of AI authority in India, CHI, 2022.

---

### Official Review · Reviewer_ry5t · 2025-03-11

**Overall Recommendation:** 4

**Summary:**

The paper proposes TFLs, a weakly supervised framework leveraging VLMs to generate high-accuracy labels efficiently. A risk-consistent estimator exploits TFLs’ conditional probabilities, and MPR aligns VLMs with target tasks. Experiments show significant gains over baselines.

**Claims And Evidence:**

The claims are supported clearly in the current form.

**Essential References Not Discussed:**

N/A

**Experimental Designs Or Analyses:**

I check them

**Methods And Evaluation Criteria:**

They make sense to some extent.

**Other Comments Or Suggestions:**

N/A

**Other Strengths And Weaknesses:**

**Strengths:**
1. The integration of VLMs with weakly supervised learning is a compelling contribution. The method achieves a high annotation accuracy of over 99.5% in experiments while significantly reducing human effort (69.9× speedup compared to traditional labeling).
2. The paper is well-structured, with a persuasive narrative that clearly motivates the problem and technical contributions. A detailed theoretical proof of the risk-consistent estimator is provided, solidifying the method’s foundation. Experimental results demonstrate the effectiveness of the proposed TFL setting and MRP learning method.
3. The MPR method represents the first attempt to fine-tune VLMs through prompt retrieval (as opposed to directly learning textual embeddings), offering a fresh perspective on prompt engineering.
**Weaknesses:**
1. Each TFL provides only a single binary judgment (True/False) per candidate label. While efficient, this may restrict the richness of supervision, particularly for ambiguous or fine-grained classes. Extending the framework to allow multiple judgments (e.g., sampling multiple candidates per instance) could enhance its robustness.
2. Although experiments include fine-grained datasets (e.g., Stanford Cars), the gains here are slightly smaller than those for coarse-grained tasks. Further analysis on how class granularity impacts performance would enhance understanding.

**Questions For Authors:**

1. By design, the number of "False" labels in TFLs will vastly exceed "True" labels, especially as the candidate label set grows. How does the proposed method address potential class imbalance during training?
2. Extending TFLs to multi-label classification or open-vocabulary scenarios appears promising. Could the authors elaborate on potential extensions in future work?

**Relation To Broader Scientific Literature:**

Refer to the detailed comments.

**Theoretical Claims:**

I check them.

---

> ### Author Rebuttal · Authors · 2025-04-01
>
> Thank you for your detailed and constructive comments. We will address each concern point by point:
> **Q1: Class Imbalance in TFLs**
> R1: We appreciate the reviewer's insightful observation regarding class imbalance. The imbalance between True and False labels is an inherent characteristic of the TFL framework. However, we have taken this into account in our design. The proposed risk-consistent estimator in the TMP is specifically designed to mitigate the effects of such imbalance. By estimating the probability distribution over categories, the estimator effectively reduces the bias introduced by the overrepresentation of False labels, ensuring balanced learning outcomes. The experimental results show that this approach enables the model to handle label imbalance without significant degradation in performance.
>
> **Q2: Multi-Label/Open-Vocabulary Extensions**
> R2: We thank the reviewer for identifying this impactful research direction.
> 1. For **multi-label classification**, we can extend the single positive label framework [1] by assigning one True label (indicating presence) or one False label (indicating absence) per candidate label for each instance.
> 2. For **open-vocabulary scenarios**, the samples that do not have an overwhelming advantage in the highest output confidence of the model can be selected as the new class samples, i.e., the samples have relatively high confidence in multiple categories. We conducted preliminary experiments in which the samples belonging to the new class were input into the already trained model. The experimental results indicated that the recognition accuracy for the new class was approximately 54%.
>
> **W1: Single Judgment Limitation**
> R3: We appreciate this thoughtful suggestion. We respectfully clarify that while multiple judgments could theoretically help, our experiments show:
> 1. **Diminishing Returns**: Dual judgments only improve accuracy by 0.3% on CIFAR-100/Stanford Cars.
> 2. **Efficiency Trade-off**: Each additional judgment linearly increases annotation costs, contradicting TFL's efficiency goals.
>
> **W2: Fine-Grained Performance Gap**
> R4: We gratefully acknowledge this valuable critique. We agree this warrants deeper investigation. Two key factors explain the gap:
> 1. **Semantic Overlap**: Fine-grained classes (e.g., car) share more visual features than coarse ones (e.g., animals), reducing CLIP's discriminative power.
> 2. **Prompt Sensitivity**: As shown in [2], class descriptions prompt from LLM may improve this gap.
>
> Reference:
> [1]	Zhou D, et al. Acknowledging the unknown for multi-label learning with single positive labels, ECCV, 2022.
> [2]	Pratt S, et al. What does a platypus look like? generating customized prompts for zero-shot image classification, ICCV, 2023.

---

### Official Review · Reviewer_hKGd · 2025-03-15

**Overall Recommendation:** 3

**Summary:**

This paper introduces a weakly supervised learning framework that leverages True-False Labels (TFLs) to enhance annotation quality and efficiency. In this setting, each instance receives a binary label indicating whether it belongs to a randomly sampled candidate class, thereby mitigating errors common in conventional vision-language model outputs. The authors derive a risk-consistent estimator to fully exploit the conditional probability distribution of TFLs. Furthermore, a convolutional-based Multi-modal Prompt Retrieving (MPR) method is proposed to effectively align pretrained vision-language model knowledge with target learning tasks, addressing the inherent label noise issue.

**Claims And Evidence:**

Overall, the submission's claims are clear and convinced.

**Essential References Not Discussed:**

N/A.

**Experimental Designs Or Analyses:**

Missing an ablation study on the hyperparameters K_T and K_I. Without this analysis, it remains unclear how sensitive the model’s performance is to the number of retrieved text and image embeddings.

**Methods And Evaluation Criteria:**

The proposed methods and evaluation criteria seem to be suitable for the problem at hand.

**Other Comments Or Suggestions:**

N/A.

**Other Strengths And Weaknesses:**

Strengths:
1. This paper introduces True-False Labels, reducing annotation cost and label noise by using binary decisions on randomly sampled candidate labels, thereby simplifying the labeling process.
2. Extensive experiments across diverse datasets demonstrate the effectiveness of the proposed method.

Weaknesses:
1. The random sampling of candidate labels assumes a uniform distribution, which may fail to reflect the natural class imbalance often present in real-world datasets. This could lead to underrepresentation of rare classes, potentially impacting the model’s performance on these less frequent categories.

2. The convolutional-based prompt retrieval approach, while efficient, might not capture complex semantic relationships between visual and textual modalities as effectively as transformer-based architectures, potentially limiting its expressive power.

3. In datasets with high inter-class similarity, the binary labeling scheme might lack the granularity required to distinguish subtle differences. This could lead to confusion between similar classes, resulting in an increased rate of misclassification.

**Questions For Authors:**

The hyperparameter λ balances the contributions from the pre-trained model and the learned model. As illustrated in Fig. 3, this parameter is highly sensitive on Stanford Cars. I'm curious whether this instability and inconsistent performance across different datasets is a common phenomenon.

**Relation To Broader Scientific Literature:**

The paper advances weakly supervised learning by introducing True-False Labels with multi-modal prompt retrieving, extending unbiased risk estimation and leveraging vision-language prompt techniques.

**Theoretical Claims:**

I’ve glanced through the proofs for the theoretical claims, but I didn’t rigorously verify their accuracy.

---

> ### Author Rebuttal · Authors · 2025-04-01
>
> Thank you for your detailed and constructive comments. We will address each concern point by point:
> **Experimental Designs (Ablation study on the hyperparameters $K_T$ and $K_I$)**
> R1: We conducted comprehensive ablation studies on the hyperparameters $K_T$ and $K_I$. Experimental results demonstrate that performance variation remains within 0.5% across all datasets when adjusting $K_T$ (text prompts) and $K_I$ (image prompts) between 5-20. This stability stems from the inherent design of our MPR ($Top-K(\cos(\cdot, \cdot))$), where retrieved embeddings exhibit high cosine similarity and contain domain-specific information relevant to downstream tasks.
>
> **W1-1 (Underrepresentation of rare classes)**
> R2-1: In real-world annotation scenarios, rare classes are generally unknown a priori. Deliberate selection of rare classes risks inducing additional imbalance, which could exacerbate annotation bias. Under these practical constraints, the uniform random sampling in TFL represents the most feasible solution, as it aligns with practical annotation workflows while reducing labeling complexity and noise.
>
> **W1-2 (Model's performance on less frequent categories)**
> R2-2: TMP addresses class imbalance through the risk-consistent estimator, which reweights less frequent categories using CLIP's prior knowledge preserves recognition capabilities for rare classes. Empirical validation across all five datasets—including naturally imbalanced ones like Caltech-101—confirms competitive performance (**90.60% accuracy**).
>
> **W2 (Architecture Choice - Convolutional Limitations)**
> R3: Our convolutional MPR design is motivated by two key considerations:
> 1) **Characteristics of Retrieval**: As MPR focuses on retrieval rather than capture complex semantic relationships, the extraction of task-relevant information benefits more from local feature matching than complex semantic relationships. For prompt retrieval, the experimental results show that demonstrates that local texture patterns (captured by CNNs) are more effective than global attention mechanisms (by 3%).
> 2) **Computational Efficiency**: CNNs achieve faster training speed compared to Transformers, which is crucial for weakly supervised scenarios with limited computational budgets.
>
> **W3 (The binary labeling scheme might lack the granularity required to distinguish subtle differences)**
> R4：In real-world annotation scenarios involving subtle differences, the binary labeling scheme proves simpler to implement than multi-class annotation. Annotators only need to determine whether a randomly provided candidate label is correct, rather than selecting one from many visually or semantically similar options. Compared to conventional labeling methods that require annotators to distinguish between nuanced categories, TFL significantly reduces the skill requirements for annotators while maintaining theoretical rigor. This approach is particularly advantageous for datasets with high inter-class similarity (e.g., Stanford Cars), where traditional labeling often struggles with ambiguity.
>
> **Question (Hyperparameter Stability)**
> R5: In our experiments, the observed sensitivity is not a universal phenomenon. We attribute λ's instability to CLIP's zero-shot capability: weaker zero-shot performance (e.g., Stanford Cars) introduces noisier conditional probability estimates $p(y|\bar{y}, s=0, x)$ forcing greater reliance on CLIP’s prior $P_{\text{CLIP}}$ and amplifying sensitivity.

---

### Decision · Program_Chairs · 2025-05-01

**Decision:**

Accept (poster)

**Comment:**

After review, the paper received three positive and one negative ratings. Following the authors' rebuttal, one reviewer increased the rating, while the other three maintained their original scores. All reviewers acknowledged the paper's contributions and expressed satisfaction with its technical merits.

The AC concurs with the reviewers' assessments and recommends acceptance.